# PGN: A POLAR GEODESIC NETWORK FOR MULTI-MODAL EMOTION RECOGNITION

## ABSTRACT

Multimodal emotion recognition faces semantic ambiguity, significant noise, and cross-modal interference including modality absence. Although psychological research supports a radial structure of emotions, many methods overlook this geometry and accumulate directional noise during fusion. We introduce the Polar Geodesic Network (PGN), which maps modality embeddings into a radial space, performs reliability-aware geodesic fusion to preserve circular topology, and then uses a Transformer to refine the fused representation and capture cross-dimensional interactions. Under a unified frozen-backbone protocol, PGN attains 0.6835 Accuracy and 0.6756 Weighted-F1 on MELD, and 0.7340 Accuracy and 0.690 Macro-F1 on IEMOCAP. Ablation results indicate that geometry-aware fusion and the subsequent Transformer contribute complementary gains. These findings demonstrate that explicit modelling in radial space enhances recognition accuracy and robustness.

## 1 INTRODUCTION

Multimodal emotion recognition integrates text, speech, and vision to improve robustness over single modalities, yet real deployments remain difficult due to semantic ambiguity near category boundaries and unreliable inputs in noisy, occluded, or out-of-distribution conditions (Ramaswamy et al., 2024; Lian et al., 2023). Surveys report that Euclidean feature fusion and standard attention often assume homogeneous reliability and linear neighbourhoods, which can cause one corrupted stream to dominate and blur decision boundaries around semantically adjacent emotions (Ramaswamy et al., 2024; Pan et al., 2023). Recent MER works emphasise robustness to missing or noisy modalities, showing that explicit modelling of modality absence or incompleteness is necessary for stable performance in practice (Lin & Hu, 2023; Wang et al., 2023). At the same time, geometry-aware learning argues that many signals live on curved manifolds where non-Euclidean distances and means preserve structure that Euclidean pooling tends to distort (Mettes et al., 2023; Tibermacine et al., 2024). For affective signals, circular or periodic representations capture directional relationships and avoid discontinuities that arise when angles are treated in a naïve Euclidean manner (Bruns et al., 2024; Tibermacine et al., 2024).

We posit that modeling multimodal affective evidence in polar coordinates, with angle for affective direction and radius for salience, and aggregating along geodesics rather than straight lines, mitigates wrap around failures where opposite directions average to neutral and enables reliability weighting to down regulate corrupted modalities during fusion (Bruns et al., 2024; Mettes et al., 2023).

**Contributions**

We propose the *Polar Geodesic Network* for geometry-aware multimodal fusion, evaluated under a standardised frozen backbone protocol with an end-to-end layer-wise learning rate decay variant for complementary analysis. We summarise three key contributions supported by recent evidence. First, we introduce a polar representation that separates phase on the circle from magnitude on the positive reals, which aligns with cyclic structure in affective variables and reduces angular discontinuities observed in Euclidean embeddings (Bruns et al., 2024; Tibermacine et al., 2024; Mettes et al., 2023). Second, we perform reliability weighted geodesic fusion by computing a circular Fréchet mean for phases and a normalised reliability average for magnitudes, which is consistent with recent findings that robust multimodal systems must explicitly handle missing or noisy modalities and that

non-Euclidean aggregation improves stability under corruption (Lin & Hu, 2023; Wang et al., 2023; Halpern et al., 2024). Third, we provide a theoretical and quantitative evaluation program that couples small noise analysis of circular estimators with standardized frozen baseline reproduction, multi seed reporting with paired testing, and controlled robustness studies for noise and missing modalities, which matches guidance from recent surveys and benchmarks on geometry aware learning and MER evaluation (Ramaswamy et al., 2024; Lian et al., 2023; Tibermacine et al., 2024).

A focused synthesis of attention based fusion, contrastive alignment, reliability modeling, geometry aware aggregation, and dialogue or graph context for MER appears in Appx. A (Ramaswamy et al., 2024; Pan et al., 2023).

## 2 PRELIMINARIES

**Notation and conventions**

Angles are measured in radians with principal representatives in $(-\pi, \pi]$. We use the centred modulus $w(x) = \mathrm{mod}_{2\pi}^{(-\pi,\pi]}(x) = ((x + \pi) \bmod 2\pi) - \pi$. On $S^1$, we adopt the complex representation $e^{i\theta}$. For phases $\{\theta_m\}$ with nonnegative weights $\{\alpha_m\}$, the (complex) resultant is

$$\mathcal{R} := \sum_m \alpha_m e^{i\theta_m} = R\, e^{i\hat{\theta}}, \qquad R = |\mathcal{R}| \in [0, 1], \ \hat{\theta} = \arg(\mathcal{R}),$$

where $R$ is the *resultant length* (polarisation) and $\hat{\theta}$ the mean direction Mardia & Jupp (2000); Jammalamadaka & SenGupta (2001); Fisher (1993). The signed angular difference is

$$\delta(\theta, \theta') = \mathrm{atan2}\big(\sin(\theta - \theta'), \cos(\theta - \theta')\big) \in (-\pi, \pi],$$

and the geodesic (shortest-arc) distance is

$$d(\theta, \theta') = |\delta(\theta, \theta')| = \min_{m \in \mathbb{Z}} |\theta - \theta' + 2\pi m|$$

(standard in circular statistics Fisher (1993); equivalent closed forms and numerical notes appear in Appendix C.1, C.2). For completeness on manifold means used later, see Pennec (2006); Afsari (2011). We use British English (e.g., normalisation, stabilise, artefacts).

### 2.1 RADIAL VS. EUCLIDEAN EMOTION REPRESENTATION

In conventional Euclidean embeddings, an emotion is represented by a vector $\mathbf{z} = (x, y)$ in which *magnitude* and *direction* are coupled. Changes in global scale (e.g., louder speech, longer text) alter $\|\mathbf{z}\|$ even when the underlying direction is unchanged, which can bias similarity towards magnitude. Although normalisation may mitigate this effect, many distance/fusion operators remain Euclidean. An illustrative discussion of scale sensitivity and the chord–arc discrepancy is provided in Appendix C.2.

We parameterise emotions in polar coordinates $(r, \theta)$, where $r$ encodes intensity/activation (or confidence) and $\theta$ encodes affective direction. This decoupling is intended to emphasise directional structure when appropriate (e.g., circumplex-like layouts Russell (1980); Posner et al. (2005); Plutchik (2001)), while retaining access to intensity via $r$. Concretely, we examine:

*(i) Reduced sensitivity to amplitude variation.* When nuisance factors primarily affect scale, angle-based comparisons can be less sensitive to such variation; formal angular distances are introduced in §2.2, and small-angle agreements with Euclidean chords are summarised in Appendix C.2. Empirical checks are reported in §5.

*(ii) Compatibility with graded/overlapping affect.* If classes occupy sector-like regions rather than isolated points, small angular mixtures vary smoothly within/between sectors, which can better capture blended states. Appendix B provides a variance-reduction analysis and a two-component ambiguity bound based on circular statistics Mardia & Jupp (2000); Jammalamadaka & SenGupta (2001); Fisher (1993).

*(iii) Invariance properties.* Angular separation is invariant to common rescalings and equivariant under global rotations (Appendix C.2); these properties aid cross-modality comparability when magnitudes are miscalibrated.

Geometry-aware operators used later (e.g., circular means/Fréchet means on $S^1$) align with classical directional statistics Mardia & Jupp (2000); Jammalamadaka & SenGupta (2001) and intrinsic means on manifolds Pennec (2006); Afsari (2011). In particular, Appendix C.3 shows that the circular Fréchet mean coincides with the resultant phase under mild dispersion, and Appendix B details variance and stability properties relevant to denoising and ambiguity tolerance. We treat the radial parameterisation as a modelling choice whose utility is validated empirically in §5.

## 2.2 Geodesic Distance

Distances on circular manifolds must respect periodicity; using plain Euclidean distance breaks this structure and causes wrap-around artefacts Mardia & Jupp (2000); Fisher (1993); Jammalamadaka & SenGupta (2001).

We adopt the notation of §2 and write the angular difference as $\Delta := \theta - \theta'$. The signed shortest angular difference is $\delta(\theta, \theta')$ and the geodesic distance is $d(\theta, \theta') = |\delta(\theta, \theta')| = \min_{m \in \mathbb{Z}} |\Delta + 2\pi m|$, with equivalent closed forms summarised in Appendix C.1 (see also Fisher (1993) for standard treatments). Geometrically, $d(\theta, \theta')$ equals the length of the *shorter arc* between two points on the unit circle Mardia & Jupp (2000). The metric axioms, boundedness $0 \le d \le \pi$, and invariances $d(\theta + 2\pi, \theta') = d(\theta, \theta')$, $d(\theta + \phi, \theta' + \phi) = d(\theta, \theta')$ are detailed in Appendix C.2 and follow standard results in circular statistics Jammalamadaka & SenGupta (2001).

The chord–arc relation is given by

$$\left| e^{i\theta} - e^{i\theta'} \right| \;=\; 2 \sin\!\left( \tfrac{1}{2} \, d(\theta, \theta') \right), \tag{2.1}$$

a classical identity on $S^1$ (Fisher, 1993, Ch. 2).

For small separations the chord approximates the geodesic distance, while near $\pi$ they diverge maximally—making chord distance unsuitable for averaging/fusion on $S^1$ Mardia & Jupp (2000). In our setting, where affect is often modelled on a circular manifold (the circumplex model), using $d$ yields similarity measures consistent with the intended topology and psychological relatedness Russell (1980); Posner et al. (2005); Plutchik (2001). For numerical stability we compute $\delta$ via $\mathrm{atan2}(\sin \Delta, \cos \Delta)$ and reproject angles to $(-\pi, \pi]$ after updates; further implementation details appear in Appendix C.2 and practical notes in Berens (2009).

## 3 Method

### 3.1 Problem Setup and Overview

We address multimodal emotion recognition with $C$ emotion classes (dataset-specific values are provided in Sec. 4). Each input sample consists of three modality streams: video $\mathcal{V}$, audio $\mathcal{A}$, and text $\mathcal{T}$. To handle varying native sampling rates, all streams are aligned to a common temporal length $T$ before fusion. The model learns an end-to-end mapping $f : (\mathcal{V}, \mathcal{A}, \mathcal{T}) \mapsto y$ where $y \in \{1, \ldots, C\}$, with all parameters—including feature extractors—jointly optimized.

#### 3.1.1 Overall Architecture

The Polar Geodesic Network (PGN) is an end-to-end framework that explicitly models the circular geometry of affective representations. As shown in Fig. 1, PGN operates through four sequential stages:

**Input encoding.** Raw modality streams are processed by trainable encoders to produce token-level embeddings, which are projected to a shared hidden dimension $H$ and common length $T$. The architecture is backbone-agnostic, supporting various encoder types (e.g., ViT/CNN for vision, self-supervised models for audio, Transformers for text).

**Polar coordinate representation (Sec. 3.2).** Each embedding dimension is decomposed into *amplitude* (intensity/confidence) and *phase* (affective direction) components. A reliability weight is estimated per modality to dynamically handle uncertain or missing inputs.

**Geodesic fusion (Sec. 3.3).** Amplitudes and phases are aggregated using weighted Fréchet means on the manifold $S^1 \times \mathbb{R}_+$, preserving circular topology and avoiding wrap-around artifacts.

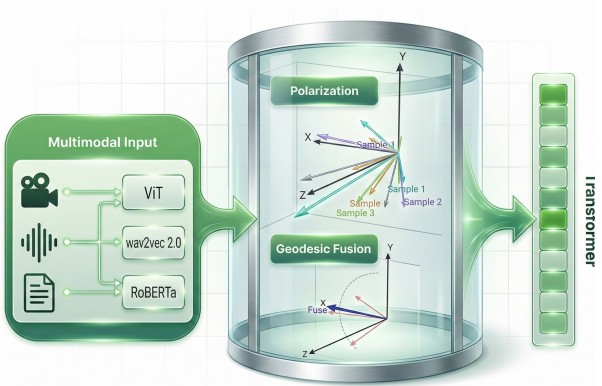

Figure 1: PGN architecture overview

Figure 1: PGN architecture overview.

**Refinement and classification (Sec. 3.4).** The fused polar representation is refined by a lightweight Transformer with geometry-aware attention, then classified into emotion categories.

**Missing modality handling.** Absent modalities are handled by setting their reliability weights to zero and renormalising across available streams, ensuring robust operation under partial observations.

## 3.2 POLAR COORDINATE REPRESENTATION

The polar coordinate representation transforms Euclidean embeddings from modality-specific encoders into a geometry-aware format that disentangles two complementary aspects of affective signals: *amplitude* (intensity) and *phase* (affective direction). By separating these components, PGN respects the circular structure of affect while providing an explicit handle for reliability weighting during fusion. This design is inspired by psychological models of emotion Russell (1980); Plutchik (2001); Posner et al. (2005) and builds on complex-valued parameterizations in geometry-aware neural networks Trabelsi et al. (2018).

### 3.2.1 AMPLITUDE AND PHASE PROJECTION

For each modality $k \in \{\text{video}, \text{audio}, \text{text}\}$ and token position $t$ (sequence length $T$), denote the encoder output by $\mathbf{x}_{k,t} \in \mathbb{R}^H$. We pass $\mathbf{x}_{k,t}$ through two lightweight MLPs (each preceded by LayerNorm Ba et al. (2016)), one to produce a nonnegative *amplitude* and the other to produce an angular *phase*:

$$\rho_{k,t} = \varepsilon_{\text{amp}} + \text{softplus}\big(\text{MLP}_\rho(\text{LayerNorm}(\mathbf{x}_{k,t}))\big), \quad (3.1)$$

$$\theta_{k,t} = \text{atan2}\big(\sin\phi_{k,t}, \cos\phi_{k,t}\big), \qquad \phi_{k,t} = \text{MLP}_\theta(\text{LayerNorm}(\mathbf{x}_{k,t})). \quad (3.2)$$

Here $\rho_{k,t} \in \mathbb{R}_+^H$ encodes tokenwise intensity per latent dimension, while $\theta_{k,t} \in (-\pi, \pi]^H$ encodes per-dimension direction on the circle. Using softplus yields smooth positive amplitudes with stable gradients Dugas et al. (2001); using $\text{atan2}(\sin, \cos)$ wraps pre-angles to principal values and preserves circular topology. For later convenience we also define the complex form $\mathbf{z}_t^{(k)} = \rho_{k,t} \odot e^{i\theta_{k,t}}$.

### 3.2.2 RELIABILITY ESTIMATION

To accommodate variable input quality, PGN parameterizes *reliability logits* at the granularity of modality $k$, token $t$, and dimension $h$ by combining amplitude strength and local phase consistency:

$$\ell_{k,t,h} = \beta_0 + \beta_\rho \rho_{k,t,h} + \beta_R R_{k,t,h}^{(\text{loc})}, \quad (3.3)$$

where $R_{k,t,h}^{(\text{loc})} \in [0, 1]$ is a local resultant-length–based consistency index computed from nearby phases (cf. §3.2). Larger amplitudes and more consistent local phases yield larger logits and hence higher normalized reliability after fusion. Cross-modality normalization is performed with a masked softmax in §3.3 using a *fixed* temperature $\tau$ (we use $\tau = 1$ unless otherwise noted). Implementation notes (the neighborhood for $R^{(\text{loc})}$ and optional temperature sensitivity) are in Appx. D.

### 3.3 GEODESIC FUSION

The geodesic fusion stage aggregates polar representations from multiple modalities on the product space $S^1 \times \mathbb{R}_+$, avoiding the topological distortions of Euclidean averaging. We rely on the distance formalism and the signed angular difference introduced in Section 2.2.

#### 3.3.1 AMPLITUDE FUSION

We normalize reliabilities *for each* $(t, h)$ across available modalities via a masked softmax with a *fixed* temperature $\tau$. Let $m_{k,t} \in \{0, 1\}$ indicate availability at time $t$ (1 if present, 0 if missing):

$$\alpha_{k,t,h} = \frac{\exp(\ell_{k,t,h}/\tau)\, m_{k,t}}{\sum_j \exp(\ell_{j,t,h}/\tau)\, m_{j,t}}, \qquad (\tau = 1 \text{ by default}). \tag{3.4}$$

This yields a strictly normalized weight distribution over the available streams (if exactly one stream is available at $t$, its weight is 1 for all $h$). Amplitude fusion on the radial factor $\mathbb{R}_+$ is then

$$\bar{\rho}_{t,h} = \sum_k \alpha_{k,t,h}\, \rho_{k,t,h}. \tag{3.5}$$

In the degenerate case where all streams are absent at $(t, h)$ (i.e., $\sum_j m_{j,t} = 0$), we set $\alpha_{\cdot,t,h} = \mathbf{0}$ and $\bar{\rho}_{t,h} = 0$; the phase term is then immaterial for downstream use.

#### 3.3.2 PHASE FUSION

On the circle $S^1$, the weighted Fréchet mean at $(t, h)$ is obtained by averaging unit complex numbers and taking the angle of the resultant. Define the (reliability–weighted) resultant vector

$$\mathbf{R}_{t,h} = \left( \sum_k \alpha_{k,t,h} \cos\theta_{k,t,h},\ \sum_k \alpha_{k,t,h} \sin\theta_{k,t,h} \right) \in \mathbb{R}^2. \tag{3.6}$$

Its direction gives the fused phase,

$$\bar{\theta}_{t,h} = \operatorname{atan2}\!\left(\mathbf{R}_{t,h}^{(y)}, \mathbf{R}_{t,h}^{(x)}\right) \in (-\pi, \pi], \tag{3.7}$$

where $\operatorname{atan2}(y, x)$ returns the principal angle of the 2D vector $(x, y)$ (correct quadrant, no division-by-zero). This equals the minimizer of the weighted sum of squared geodesic distances (proof in Appx. C.3).

The *resultant length* is the Euclidean norm of $\mathbf{R}_{t,h}$,

$$R_{t,h} = \|\mathbf{R}_{t,h}\|_2 = \sqrt{ \left( \sum_k \alpha_{k,t,h} \cos\theta_{k,t,h} \right)^2 + \left( \sum_k \alpha_{k,t,h} \sin\theta_{k,t,h} \right)^2 } \in [0, 1], \tag{3.8}$$

which quantifies the agreement (concentration) among phases: larger $R_{t,h}$ indicates stronger consensus. Properties and bounds of $R_{t,h}$ are summarized in Appx. B. This geometry-aware fusion avoids the wrap-around artifacts discussed in Section 2.2.

#### 3.3.3 GRADIENT STABILITY

We stabilize training with three ingredients:
(i) *Geometry-aligned differences.* We use the signed, wrapped angular difference and the post-update phase reprojection defined in §2.2 to avoid branch-cut discontinuities.
(ii) *Uncertain-phase damping.* When the resultant length $R_{t,h}$ is small (high-variance phases; see Appx. B), we gate *phase-side* gradients by a factor $g_{t,h} \in [0, 1]$ increasing in $R_{t,h}$, e.g.

$$g_{t,h} = R_{t,h} \qquad \text{or} \qquad g_{t,h} = R_{t,h}^{\gamma} \ (\gamma \geq 1), \tag{3.9}$$

so updates focus on amplitude/reliability until angular evidence becomes reliable.
(iii) *Safe numerics.* We clamp resultant lengths where they appear in denominators and apply global-norm gradient clipping:

$$R_{t,h} \leftarrow \max\!\left(R_{t,h}, \varepsilon_{\mathrm{amp}}\right), \qquad \|\nabla\Theta\| \leftarrow \min\left(\|\nabla\Theta\|, \mathrm{clip}\right). \tag{3.10}$$

Finally, the fused complex representation passed to refinement is

$$z_{t,h} = \bar{\rho}_{t,h}\, e^{i\bar{\theta}_{t,h}}, \qquad u_t = \left[ \Re(z_t);\ \Im(z_t) \right] \in \mathbb{R}^{2H}. \tag{3.11}$$

### 3.4 LEARNING PROCEDURE WITH TRANSFORMER REFINEMENT

PGN integrates geometry-aware fusion with neural attention and is trained end-to-end. After geodesic fusion (§3.3), we take the realified per-token inputs $\{\mathbf{u}_t\}_{t=1}^T \in \mathbb{R}^{2H}$ (constructed from $\mathbf{z}_t = \bar{\rho}_t \odot e^{i\bar{\theta}_t}$) and feed them to a lightweight Transformer Vaswani et al. (2017). We augment self-attention logits with a geometry-aware bias that prefers tokens with strong amplitudes and aligned phases:

$$A_{ij} = \frac{\mathbf{q}_i^\top \mathbf{k}_j}{\sqrt{d}} + \lambda_g\, G_{ij}, \qquad G_{ij} = \tfrac{1}{H}\, \Re(\mathbf{z}_i^* \cdot \mathbf{z}_j).$$

A concise, end-to-end view of PGN's computation and learning loop is given in Algorithm 1. Implementation details (QKV projections, sharing of $\lambda_g$, complexity) are in Appx. E.

**Objective and optimisation**

We minimise a composite objective combining cross-entropy with two geometry-aligned regularisers—reliability entropy (to discourage single-modality collapse) and phase diversity (to avoid angular collapse)—plus weight decay. Optimisation uses AdamW Loshchilov & Hutter (2017) with cosine decay, linear warmup, and global-norm clipping ($c=1.0$). Exact formulas and hyperparameters are in Appx. F. A progressive schedule (warm-up $\rightarrow$ partial unfreezing $\rightarrow$ full joint training) further stabilises training; see Appx. F.

---

**Algorithm 1** PGN: End-to-end Geodesic Fusion with Geometric-Refined Transformer (mini-batch)

---

1: **Input:** streams $(\mathcal{V}, \mathcal{A}, \mathcal{T})$ with availability masks $m_{k,t} \in \{0,1\}$; batch size $B$; parameters $\Theta$.
2: Encode $\rightarrow$ Project/align: $\tilde{x}_k \in \mathbb{R}^{B \times T \times H}$                                                         (§3.1)
3: Polar heads (per token/dim): amplitude by (3.1), phase wrapping by (3.2)         (§3.2)
4: Local consistency $R^{(\mathrm{loc})}$                                                            (Appx. C)
5: Reliability logits by (3.3)                                                              (§3.2)
6: Masked softmax (fixed $\tau$) by (3.4)                                 (§3.3, Appx. C)
7: Fuse amplitude by (3.5)                                                        (§3.3)
8: Resultant & fused phase by (3.6)–(3.7); resultant length by (3.8)    (§3.3; Appx. B.3)
9: Stability: phase-grad gate by (3.9); clamp/clip by (3.10)                   (§3.3)
10: Complex $\rightarrow$ real: by (3.11)                                              (§3.3)
11: Geom bias & attention: use $G_{ij}$ and $A_{ij}$                       (§3.4; Appx. D)
12: Loss & update                                                       (Appx. E)

---

## 4 EXPERIMENTS

### 4.1 DATASETS & BASELINES

We evaluate categorical emotion recognition on MELD Poria et al. (2019) and IEMOCAP Busso et al. (2008). A complementary sentiment benchmark on MOSEI Zadeh et al. (2018) is provided in the appendix for reference only.

To isolate the contribution of our polar–geodesic fusion (PGN) and ensure strict comparability, we adopt a frozen-backbone setting by default: all encoders (text, audio, vision) are fixed across methods and trained heads/fusion modules share the *same* preprocessing, training budget, and seeds.

Under this unified frozen protocol we *reproduce* strong multimodal fusion baselines, including *MulT* Tsai et al. (2019), *MemoCMT* Author list per journal PDF (2025), *MultiEMO* Shi & Huang (2023), and the calibration head of *CMERC* Tu et al. (2024b). These constitute the principal baselines used in our frozen-backbone SOTA tables on MELD and IEMOCAP. All reproduced baselines operate strictly under the same encoder features and differ *only* in their fusion/head architecture, enabling a controlled comparison of modeling choices.

In addition, we report an *end-to-end* PGN variant with layer-wise learning-rate decay (LLRD) under the *same* update/epoch budget; this E2E variant is compared *only* against our frozen PGN in the appendix and is not used for any claim against literature-only methods.

For completeness, we also include a separate *produced-results* comparison in which PGN (E2E) is compared to recent system-level results reported in the literature. On MELD, these include MultiEMO Shi & Huang (2023), CMERC Tu et al. (2024b), AdaIGN Tu et al. (2024a), McDiff Chen et al. (2023), and MemoCMT Author list per journal PDF (2025). On IEMOCAP, we use the same set of system-level baselines, and additionally discuss HARDY-MER as a robustness-oriented reference model in our missing-modality experiments. These literature numbers are never mixed with reproduced results, and no statistical testing is performed against them.

Finally, in our missing-modality experiments on IEMOCAP and MOSEI, we directly compare PGN with HARDY-MER, a state-of-the-art model explicitly designed for missing-modality robustness. These results are reported in a dedicated robustness section and are *not* used in the frozen-backbone comparisons.

**Metrics and evaluation protocol.**

On MELD we follow standard practice and report *Weighted-F1*, *Macro-F1*, and *Accuracy*. On IEMOCAP we report *Accuracy* and *Macro-F1*. Per-class metrics (P/R/F1) and confusion matrices are provided in the appendix to highlight error structure.

All scores are reported as *mean$\pm$std* over a fixed seed set (five seeds unless noted). Unless specified otherwise, we train with AdamW for 50 epochs. Validation is performed each epoch; we evaluate the single best-on-validation checkpoint per seed on the test split.

Significance is assessed with paired $t$-tests across seeds and *restricted to reproduced (frozen) runs* (PGN vs. MulT/MemoCMT/MultiEMO/CMERC). We additionally report effect sizes (Cohen's $d$) in the supplement when relevant. We do not test against literature-only numbers, and we do not mix metrics across sources.

### 4.2 ABLATIONS AND ORDER SENSITIVITY

Ablations are run under the same frozen protocol on both datasets (MELD and IEMOCAP) to quantify the contribution of geodesic fusion and the complex transformer. We also probe module order (PGT vs. PTG) and report the resulting deltas (accuracy and F1 score) to establish the importance of geometry-first alignment.

We apply controlled stressors on MELD and IEMOCAP: audio noise at 20/10/5 dB SNR, video occlusions at 10%/30%, text noise at 5%/10%, single/dual modality missing (A/V/T; A+V), and random token/frame drop with $p \in \{0.1, 0.3, 0.5\}$. For each condition we report the same primary metrics as in the clean setting and include absolute/relative drops. Efficiency measurements (latency, peak memory, train time) use the same hardware and pipeline across methods. Additional details—label mappings, seed list, bootstrap and perturbation specs, and the MOSEI reference benchmark (MISA/M3ER only)—are provided in Appx. G.

## 5 MAIN RESULTS

### 5.1 COMPARISON WITH STATE-OF-THE-ART METHODS

We evaluate PGN under two complementary settings: (i) a unified *frozen-backbone* protocol, where all baselines share the same encoders, preprocessing, training budget, and fixed seeds; and (ii) a separate *produced-results* comparison against reported system-level numbers from the literature. The frozen setting isolates the fusion/head architecture, while the produced-results tables situate PGN among recent multimodal ERC systems.

All frozen-backbone experiments use a 50-epoch budget with per-epoch validation; test performance is taken from the best-on-validation checkpoint per seed.

*Frozen-backbone comparison (MELD and IEMOCAP).* Across both datasets, PGN achieves large and consistent improvements over reproduced transformer/attention baselines. On MELD, gains appear simultaneously on Accuracy, Macro-F1, and Weighted-F1 with small seed variance, indicating stable behaviour and suggesting that the polar–geodesic parameterization improves both correctness

and class balance. On IEMOCAP, PGN also leads by a substantial margin in Accuracy and Macro-F1, with improvements distributed across emotions rather than concentrated on dominant classes.

Table 1: **Frozen-backbone** comparison on MELD and IEMOCAP (mean±std over fixed seeds).

| Method | MELD | | | IEMOCAP | |
|---|---|---|---|---|---|
| | **Acc** | **Macro-F1** | **Weighted-F1** | **Acc** | **Macro-F1** |
| PGN (ours) ‡ | **0.6835** [± 0.006] | **0.5953** [± 0.008] | **0.6756** [± 0.007] | **0.7340** [± 0.012] | **0.690** [± 0.015] |
| MemoCMT ‡ | 0.5761 [± 0.009] | 0.4184 [± 0.011] | 0.5365 [± 0.010] | 0.582 [± 0.018] | 0.534 [± 0.017] |
| MulT ‡ | 0.5389 [± 0.010] | 0.3973 [± 0.010] | 0.4575 [± 0.012] | 0.571 [± 0.016] | 0.512 [± 0.014] |
| MultiEMO ‡ | 0.6120 [± 0.012] | 0.5050 [± 0.014] | 0.6420 [± 0.015] | 0.631 [± 0.013] | 0.582 [± 0.016] |
| CMERC-head ‡ | 0.6013 [± 0.011] | 0.4872 [± 0.013] | 0.6308 [± 0.012] | 0.612 [± 0.015] | 0.566 [± 0.014] |

*Legend:* ‡ reproduced (frozen; unified setup). *Primary:* Accuracy and Macro-F1.

**Comparison to reported SOTA results.**

To contextualize PGN as a full system, we also compare the *end-to-end PGN* model (with LLRD) to recent system-level ERC models using the authors' reported numbers. These comparisons are separate from the frozen-backbone analyses and involve no statistical testing against reproduced baselines.

Table 2: **Reported results** on MELD and IEMOCAP.

| Method | MELD (reported) | | IEMOCAP (reported) | |
|---|---|---|---|---|
| | **Acc** | **Weighted-F1** | **Acc** | **Weighted-F1** |
| **PGN (ours)** | **0.6869** | **0.6790** | **0.7377** | 0.6930 |
| MultiEMO Shi & Huang (2023) | – | 0.6674 | – | 0.7284 |
| AdaIGN (IGN) Tu et al. (2024a) | 0.6762 | 0.6679 | 0.7049 | 0.7074 |
| CMERC Tu et al. (2024b) | – | 0.6685 | – | 0.7198 |
| M2FNet Chudasama et al. (2022) | 0.6785 | 0.6671 | 0.6969 | 0.6986 |
| MM-DFN Hu et al. (2022) | 0.6249 | 0.5946 | 0.6821 | 0.6818 |
| MMGCN Hu et al. (2021) | – | 0.5865 | – | 0.6622 |
| Conversation MER Li et al. (2025) | 0.6880 | 0.6780 | 0.7190 | 0.7240 |

To understand error structure, Figure 2 shows normalized confusion matrices for PGN. Rows sum to one per true class; typical confusion pairs (*sad → neutral* on MELD; *angry → frustrated* on IEMO-CAP) are reduced, indicating improved separation of adjacent affective states under geometry-first fusion. In addtion, the Qualitative structure of the learned radial space is provided in Appendix H.2, where we visualise utterance-level polar embeddings on MELD and IEMOCAP. These plots show that PGN organises emotions into coarse angular sectors with radius reflecting confidence, consistent with the geometry introduced in Sec. 3.

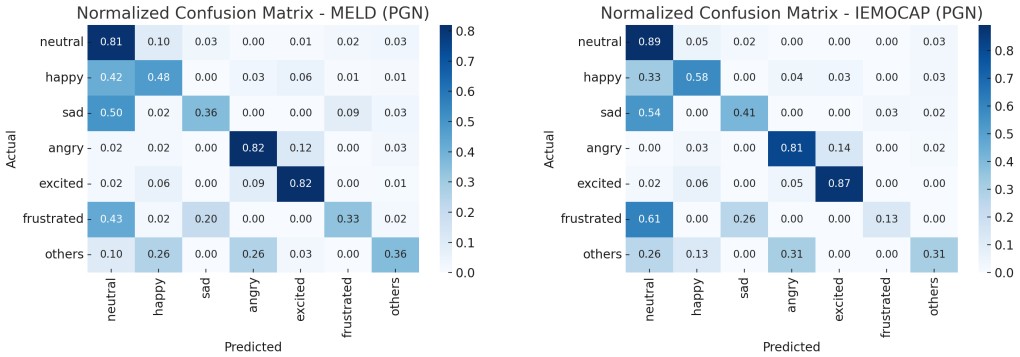

Figure 2: Normalized confusion matrices for PGN (test). Left: **MELD**. Right: **IEMOCAP**.

Rows normalized to sum to 1; reduced mass on typical confusion pairs indicates improved separation across adjacent classes.

## 5.2 ABLATION STUDY

Full ablation results are provided in Appendix I, but we summarize the key findings here. Removing geodesic fusion consistently degrades performance across both MELD and IEMOCAP, indicating that geometry reduces phase variance and stabilizes cross-modal alignment before attention. Dropping the complex transformer leads to the largest performance drops, showing that cross-modal dependencies not captured by geometric alignment are still crucial. We also examine module order and find that applying geometry-first alignment (PGT) yields steady gains, whereas reversing the order (PTG) incurs substantial penalties on both datasets, confirming that attention is most effective after geometric normalization.

## 5.3 ROBUSTNESS AND EFFICIENCY

We evaluate robustness to audio SNR and random token/frame drop, and summarize efficiency. PGN maintains higher curves under degradations while retaining competitive latency and memory in the frozen setting, indicating that geometry-first fusion reduces destructive averaging without heavier heads.

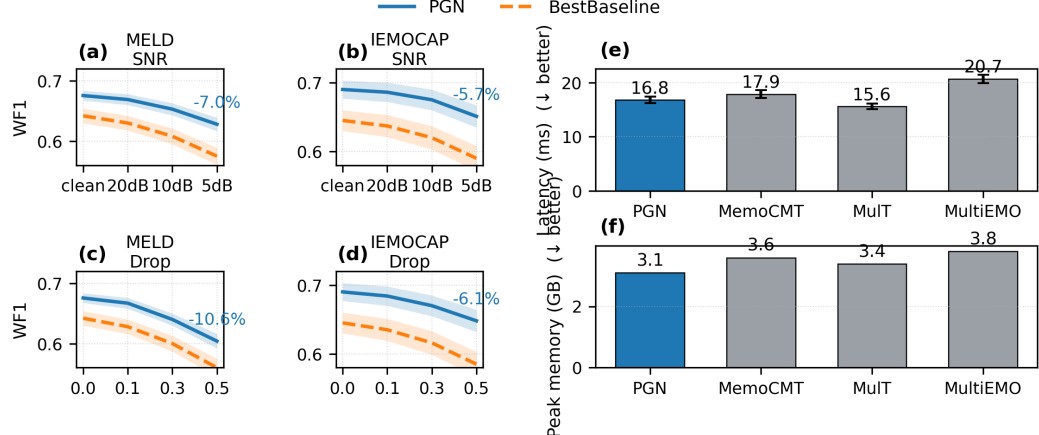

Figure 3: Robustness and efficiency overview.

**Left (a–d):** sensitivity under **audio SNR** {clean, 20, 10, 5 dB} and **random drop** $p \in \{0, 0.1, 0.3, 0.5\}$. MELD reports *Weighted-F1*; IEMOCAP reports *Macro-F1*. Bands show mean±std across seeds.
**Right (e–f): inference latency** (median of 100 inference runs; 20-run warmup) and **peak memory** under identical hardware/batch.

## 6 CONCLUSION

This research introduces a geometric view of multimodal emotion recognition by polarising features and fusing them through geodesic alignment. PGN provides a simple yet principled mechanism for handling cross-modal discrepancies and the ambiguity inherent in affective signals. Our experiments reflect this: under a strictly controlled frozen-backbone setting, PGN consistently improves over reproduced baselines, indicating that its gains stem from the fusion mechanism rather than model size or training heuristics. In end-to-end comparisons with reported results, PGN remains competitive despite using lighter encoders, suggesting that geometric alignment is broadly effective across settings. A natural next step is to evaluate PGN under larger and more diverse conditions to better understand when geometric structure offers the strongest generalization benefits in multimodal emotion recognition and how such geometry can better capture the natural distribution of affective states.

**Acknowledgments**

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

## A  RELATED WORK

**Surveys and task framing**

Recent surveys synthesise datasets, features, and fusion strategies for multimodal emotion recognition, consistently highlighting challenges from semantic ambiguity and input unreliability in realistic conditions (Ramaswamy et al., 2024; Lian et al., 2023; Pan et al., 2023). These reviews also emphasise that many pipelines still rely on Euclidean representations and simple pooling or attention on top of Euclidean features, which can implicitly assume homogeneous reliability across modalities (Ramaswamy et al., 2024; Pan et al., 2023).

**Feature aggregation and attention-based fusion**

Classical early and late fusion are computationally simple yet struggle with heterogeneous reliability and cross-modal alignment, motivating attention-based designs that adaptively weight salient cues (Lian et al., 2023; Pan et al., 2023). Surveyed evidence indicates that attention helps capture cross-modal dependencies but may still operate in a linear neighbourhood assumption that is brittle when one stream is corrupted or when labels are ambiguous near class boundaries (Ramaswamy et al., 2024; Pan et al., 2023).

**Robustness to missing or noisy modalities**

A growing line of MER work targets robustness by explicitly handling modality absence or corruption, improving stability when inputs are incomplete or degraded (Lin & Hu, 2023; Wang et al., 2023). Complementary approaches modulate features to cope with missing signals and report gains under controlled ablation of modalities, reinforcing the need for reliability-aware fusion (Halpern et al., 2024; Lin & Hu, 2023).

**Geometry-aware representations and aggregation**

Non-Euclidean learning argues that many signals lie on curved manifolds and that geodesic distances and means preserve structure better than Euclidean pooling, which can distort cyclic or hierarchical relations (Mettes et al., 2023; Tibermacine et al., 2024). For circular variables, deep circular regression and Riemannian treatments avoid discontinuities at angle wrap-around and support estimators that better respect directional similarity, motivating polar encodings in affective settings (Bruns et al., 2024; Tibermacine et al., 2024).

**Positioning of our approach**

The above strands suggest three design needs for MER in the wild: an affect-consistent latent geometry to reduce wrap-around artifacts, a reliability-aware fusion rule to resist noisy or missing inputs, and a standardized evaluation protocol that disentangles architectural gains from backbone tuning (Ramaswamy et al., 2024; Lian et al., 2023). Our work connects these needs by using a polar representation and geodesic aggregation that align with circular affect structure, together with a reliability-weighted fusion recipe and a frozen-backbone evaluation backed by multi-seed statistics and controlled robustness tests (Mettes et al., 2023; Lin & Hu, 2023).

## B  POLARISATION OF PHASES: RESULTANT AS A CONSISTENCY

**Setup**

Let angles $\{\theta_m\}_{m=1}^M$ with nonnegative weights $\{\alpha_m\}$, $\sum_m \alpha_m = 1$. Define the complex resultant

$$R\,e^{i\hat{\theta}} \;=\; \sum_{m=1}^M \alpha_m\,e^{i\theta_m}, \qquad R \in [0,1],\ \hat{\theta} \in (-\pi, \pi].$$

Here $R = \|\mathbf{R}\|$ measures *polarisation* (agreement) and $\hat{\theta}$ the fused phase.

**Polarisation identity (pairwise form)**

**Proposition 1** (Polarisation identity). *The resultant length admits the exact decomposition*

$$R^2 \;=\; \sum_{m=1}^M \alpha_m^2 \;+\; 2 \sum_{1 \le i < j \le M} \alpha_i \alpha_j \cos(\theta_i - \theta_j).$$

*Proof.* Write $Re^{i\hat{\theta}} = \sum_m \alpha_m e^{i\theta_m}$ and take the squared modulus: $R^2 = \left|\sum_m \alpha_m e^{i\theta_m}\right|^2 = \sum_m \alpha_m^2 + \sum_{i \ne j} \alpha_i \alpha_j e^{i(\theta_i - \theta_j)}$. Taking the real part yields the stated form. $\qquad\square$

**Immediate corollaries**

**Corollary 1** (Bounds and equality cases). $0 \le R \le 1$. *Moreover, $R = 1$ iff all $\theta_m$ are identical (perfect alignment); $R = 0$ is attainable under antipodal cancellation (e.g., two opposite directions with equal total weight).*

**Corollary 2** (Monotonicity w.r.t. dispersion). *If all pairwise separations shrink (i.e., each $\cos(\theta_i - \theta_j)$ weakly increases), then $R$ weakly increases by Prop. 1. Thus $R$ is a* consistency *index.*

**Corollary 3** (Lower bound by dominant weight). *Let $\alpha_{\max} = \max_m \alpha_m$. Then*

$$R \;\ge\; \max\{0,\; 2\alpha_{\max} - 1\}.$$

*In particular, if one modality dominates ($\alpha_{\max} \ge \frac{1}{2}$), the resultant cannot vanish.*

*Sketch.* Group all non-dominant phases adversarially against the dominant one; the worst-case is an antipodal placement, yielding the stated bound from vector subtraction geometry. $\qquad\square$

**Link to denoising and ambiguity tolerance**

**Proposition 2** (Variance–polarisation coupling). *Under small independent angular noises with variances $\{\sigma_m^2\}$ and weights $\{\alpha_m\}$, the fused phase satisfies the delta-method approximation*

$$\mathrm{Var}(\hat{\theta}) \;\approx\; \frac{\sum_m \alpha_m^2 \sigma_m^2}{R^2},$$

*so larger polarisation $R$ yields smaller angular variance (denoising).*

*Sketch.* Linearise the map $\{\theta_m\} \mapsto \hat{\theta} = \mathrm{atan2}(R_y, R_x)$ around the noise-free configuration; the Jacobian has norm proportional to $1/R$. See also classical results in circular statistics (Mardia & Jupp, 2000; Jammalamadaka & SenGupta, 2001; Fisher, 1993). $\qquad\square$

**Proposition 3** (Two-component ambiguity bound). *For two components at $\theta_a, \theta_b$ with weights $\alpha, 1-\alpha$ and gap $\delta = |\theta_a - \theta_b| \le \pi$,*

$$\left| \hat{\theta} - \tfrac{1}{2} w(\theta_a + \theta_b) \right| \;\le\; \left| \alpha - \tfrac{1}{2} \right| \cdot \delta, \qquad R \;=\; \sqrt{\alpha^2 + (1-\alpha)^2 + 2\alpha(1-\alpha)\cos\delta}.$$

*Hence for small $\delta$ and near-balanced weights, the fused angle deviates by $O(\delta)$ and $R$ remains high (smooth tolerance to ambiguity/co-occurrence).*

**Remarks**

(i) Multi-modal or near-antipodal configurations can yield small $R$ and multiple circular means; robust weighting (e.g., down-weighting uncertain modalities) mitigates this. (ii) Propositions 2–3 connect $R$ to uncertainty: $R$ is a natural confidence proxy used by PGN for weighting (cf. Method, §2.2). (iii) See also intrinsic mean existence/uniqueness conditions on manifolds (Pennec, 2006; Afsari, 2011).

## C   GEODESIC DISTANCE: CLOSED FORMS AND PROOFS

### C.1   EQUIVALENT CLOSED FORMS ON $S^1$

**Setup**

Angles are in radians with principal representatives $\theta, \theta' \in (-\pi, \pi]$ and $\Delta = \theta - \theta'$.

**Centred modulus (explicit)**

We map any $x \in \mathbb{R}$ to its principal representative in $(-\pi, \pi]$ via

$$\mod_{2\pi}^{(-\pi,\pi]}(x) \;=\; \big((x + \pi) \bmod 2\pi\big) - \pi,$$

and write $w(x) := \mod_{2\pi}^{(-\pi,\pi]}(x) \in (-\pi, \pi]$. We adopt this convention throughout the paper.

**From minimisation to centred modulus**

With $d(\theta, \theta') = \min_{m \in \mathbb{Z}} |\Delta + 2\pi m|$, the minimising $m$ is the one that maps $\Delta$ into $(-\pi, \pi]$, hence

$$d(\theta, \theta') \;=\; |w(\Delta)|.$$

**Absolute-value composition form**

For any $x \in \mathbb{R}$, $|w(x)| = \pi - \big|\pi - |x|\big| \in [0, \pi]$, yielding the compact closed form used in the main text:

$$d(\theta, \theta') \;=\; \pi - \Big|\pi - \big|\Delta\big|\Big|.$$

**Signed difference equivalence**

Let $\delta(\theta, \theta') = \operatorname{atan2}(\sin \Delta, \cos \Delta) \in (-\pi, \pi]$. Since $(\cos \Delta, \sin \Delta)$ lies on the unit circle at angle $w(\Delta)$, we have

$$\delta(\theta, \theta') \;=\; w(\Delta), \qquad d(\theta, \theta') \;=\; |\delta(\theta, \theta')|.$$

**Chord–arc identity (proof of Eq. 2.1)**

For unit vectors with angles $\theta, \theta'$,

$$\|\mathbf{u} - \mathbf{v}\|_2^2 \;=\; 2 - 2\cos d(\theta, \theta') \;=\; 4\sin^2\!\Big(\tfrac{1}{2}d(\theta, \theta')\Big),$$

hence $\|\mathbf{u} - \mathbf{v}\|_2 = 2\sin\!\big(\tfrac{1}{2}d(\theta, \theta')\big)$.

### C.2   METRIC PROPERTIES AND NUMERICAL NOTES

**Metric axioms, periodicity, boundedness, rotation invariance**

With $d(\theta, \theta') = |w(\theta - \theta')|$ and $w = \mod_{2\pi}^{(-\pi,\pi]}$, non-negativity, identity, and symmetry are immediate. Periodicity, boundedness, and rotation invariance read:

$$d(\theta + 2\pi m, \theta') \;=\; d(\theta, \theta') \quad \text{for all } m \in \mathbb{Z}, \tag{C.1}$$

$$0 \;\leq\; d(\theta, \theta') \;\leq\; \pi, \tag{C.2}$$

$$d(\theta + \phi, \theta' + \phi) \;=\; d(\theta, \theta') \quad \text{for all } \phi \in \mathbb{R}. \tag{C.3}$$

**Triangle inequality (quotient-space proof)**

Let $\mathbb{T} = \mathbb{R}/(2\pi\mathbb{Z})$ with metric

$$d(\bar{x}, \bar{y}) \;=\; \inf_{m \in \mathbb{Z}} |x - y + 2\pi m|. \tag{C.4}$$

For $a = \theta_a - \theta_b$ and $b = \theta_b - \theta_c$, we have

$$d(\theta_a, \theta_c) \;=\; \inf_{m \in \mathbb{Z}} |(a+b) + 2\pi m| \;\leq\; \inf_{m \in \mathbb{Z}} |a + 2\pi m| \;+\; \inf_{n \in \mathbb{Z}} |b + 2\pi n| \;=\; d(\theta_a, \theta_b) + d(\theta_b, \theta_c),$$
$$\tag{C.5}$$

an application of the standard inf-convolution argument on the quotient metric.

**Small-angle consistency; Lipschitz bounds; gradients; numerics**

From Eq. 2.1,

$$\|\mathbf{u} - \mathbf{v}\|_2 \;=\; 2\sin\!\left(\tfrac{1}{2}\, d(\theta, \theta')\right) \sim d(\theta, \theta') \quad \text{as } d \to 0,$$

whereas near $\pi$ they diverge maximally.

*Separate 1-Lipschitz in each argument.* Since $w(\cdot)$ is 1-Lipschitz on $\mathbb{R}$ under the centred modulus,

$$|d(\theta + \varepsilon, \theta') - d(\theta, \theta')| \leq |\varepsilon|, \qquad |d(\theta, \theta' + \varepsilon) - d(\theta, \theta')| \leq |\varepsilon|.$$

*Joint 1-Lipschitz w.r.t. $L^1$.* For any $(\theta, \theta'), (\phi, \phi')$,

$$|\, d(\theta, \theta') - d(\phi, \phi')\,| \;\leq\; |\theta - \phi| + |\theta' - \phi'|.$$

*Gradients of the signed angle.* Let $\Delta = \theta - \theta'$, $c = \cos\Delta$, $s = \sin\Delta$, and $\delta(\theta, \theta') = \operatorname{atan2}(s, c) \in (-\pi, \pi]$. Away from wrap points $\Delta \in \{\pm\pi\}$ and configurations where the resultant magnitude vanishes (near-antipodal cancellations; a measure-zero set under generic perturbations),

$$\frac{\partial \delta}{\partial \Delta} = 1, \qquad \frac{\partial \delta}{\partial \theta} = +1, \qquad \frac{\partial \delta}{\partial \theta'} = -1.$$

Thus $\nabla |\delta|$ is well-defined almost everywhere; at wraps, use subgradients or add a small jitter.

*Numerical notes* (i) Prefer `atan2(s,c)` to $\arctan(s/c)$ to avoid division by zero and obtain the correct quadrant; (ii) when aggregating multiple phases, guard against near cancellation by adding a tiny $\varepsilon$ to the resultant magnitude before normalisation; (iii) after updates, reproject angles to $(-\pi, \pi]$ using $w(x) = ((x + \pi) \bmod 2\pi) - \pi$.

### C.3 CIRCULAR FRÉCHET MEAN EQUALS RESULTANT PHASE

**Statement**

Given angles $\{\theta_m\}_{m=1}^M$ with nonnegative weights $\{\alpha_m\}$, define

$$\mathbf{R} = \Big( \sum_m \alpha_m \cos\theta_m, \; \sum_m \alpha_m \sin\theta_m \Big), \quad R = \|\mathbf{R}\|, \quad \theta^\star = \operatorname{atan2}(R_y, R_x).$$

If the weighted sample is not antipodally symmetric and has nonzero resultant length $R > 0$ (i.e., no exact cancellation), then $\theta^\star$ is the unique minimiser of

$$\theta \;\mapsto\; \sum_{m=1}^M \alpha_m \, d(\theta, \theta_m)^2,$$

i.e., the circular Fréchet mean equals the angle of the resultant.

**Proof sketch**

Using $\delta(\theta, \theta_m)$,

$$\sum_m \alpha_m\, \delta(\theta, \theta_m)^2 \;=\; \sum_m \alpha_m \big(\theta - \theta_m\big)^2 \quad \text{modulo } 2\pi \text{ wraps}.$$

Differentiating w.r.t. $\theta$ (ignoring wrap points) yields the first-order condition $\sum_m \alpha_m \sin(\theta - \theta_m) = 0$ and second-order positivity $\sum_m \alpha_m \cos(\theta - \theta_m) > 0$ under mild dispersion. These give

$$\tan\theta^\star = \frac{\sum_m \alpha_m \sin\theta_m}{\sum_m \alpha_m \cos\theta_m} \;\Rightarrow\; \theta^\star = \operatorname{atan2}\Big( \sum_m \alpha_m \sin\theta_m, \; \sum_m \alpha_m \cos\theta_m \Big).$$

**Remarks**

(i) Multiple means may exist when $R = 0$ or mass concentrates at opposite directions; in such cases, regularisation or initialisation near the resultant phase can help. (ii) In our experiments, datasets and weightings satisfy the mild-dispersion condition almost everywhere.

# D ALGORITHMIC DETAILS FOR POLAR PROJECTION AND RELIABILITY

**Scope and notation**

This appendix specifies the algorithmic details for the polar projection and reliability estimation used in §3.2–3.3. For modality $k \in \{\text{video}, \text{audio}, \text{text}\}$, time $t \in \{1, \ldots, T\}$, and dimension $h \in \{1, \ldots, H\}$, the encoder output is $\mathbf{x}_{k,t} \in \mathbb{R}^H$. All operations below are per-token/per-dimension unless stated otherwise; $\odot$ denotes elementwise multiplication. We write

$$\text{wrap}(\phi) = \text{atan2}(\sin \phi, \cos \phi) \in (-\pi, \pi].$$

**Parameter sharing**

The amplitude head is *shared across modalities*, while the phase head is *modality-specific*. Concretely, $(\mathbf{W}_{\rho,1}, \mathbf{b}_{\rho,1}, \mathbf{W}_{\rho,2}, \mathbf{b}_{\rho,2})$ are shared for all $k$, whereas $(\mathbf{W}_{\theta,1}^{(k)}, \mathbf{b}_{\theta,1}^{(k)}, \mathbf{W}_{\theta,2}^{(k)}, \mathbf{b}_{\theta,2}^{(k)})$ are per-modality parameters.

**Shapes and LayerNorm**

Unless otherwise noted, $\mathbf{W}_{\rho,1}, \mathbf{W}_{\theta,1} \in \mathbb{R}^{H \times H}$, $\mathbf{b}_{\rho,1}, \mathbf{b}_{\theta,1} \in \mathbb{R}^H$, and analogously for the second layers. LayerNorm is applied over the feature dimension $H$.

**Polar projection (recap)**

As in §3.2, amplitudes and phases are obtained via two lightweight MLPs:

$$\mathbf{h}_{\rho,k,t} = \text{ReLU}\big(\mathbf{W}_{\rho,1} \, \text{LayerNorm}(\mathbf{x}_{k,t}) + \mathbf{b}_{\rho,1}\big), \quad \rho_{k,t} = \varepsilon_{\text{amp}} + \text{softplus}\big(\mathbf{W}_{\rho,2} \mathbf{h}_{\rho,k,t} + \mathbf{b}_{\rho,2}\big),$$

$$\mathbf{h}_{\theta,k,t} = \text{ReLU}\big(\mathbf{W}_{\theta,1} \, \text{LayerNorm}(\mathbf{x}_{k,t}) + \mathbf{b}_{\theta,1}\big), \quad \phi_{k,t} = \mathbf{W}_{\theta,2} \mathbf{h}_{\theta,k,t} + \mathbf{b}_{\theta,2},$$

$$\theta_{k,t} = \text{wrap}(\phi_{k,t}) \in (-\pi, \pi]^H.$$

We use a small amplitude floor $\varepsilon_{\text{amp}} > 0$ (e.g., $10^{-6}$) to keep amplitudes away from zero.

**Local phase consistency**

For robustness to local angular noise, we compute a per-$(k, t, h)$ consistency index as the resultant length over a small temporal neighborhood $\mathcal{N}(t)$:

$$R_{k,t,h}^{(\text{loc})} = \left\| \frac{1}{|\mathcal{N}(t)|} \sum_{\tau \in \mathcal{N}(t)} \exp\big(i \, \theta_{k,\tau,h}\big) \right\| \in [0, 1],$$

where $\mathcal{N}(t)$ is a radius-$r$ window (causal or noncausal) with boundary indices clipped to $[1, T]$. Uniform averaging is used by default; a tapered kernel (triangular/Gaussian) yields similar behavior. Interpretation and bounds follow Appx. B.

**Reliability logits**

Reliability is parameterized by logits that combine amplitude strength and local phase consistency:

$$\ell_{k,t,h} = \beta_0 + \beta_\rho \, \rho_{k,t,h} + \beta_R \, R_{k,t,h}^{(\text{loc})},$$

with scalars $(\beta_0, \beta_\rho, \beta_R)$ learned jointly with the model.

**Masked-softmax normalization (fixed temperature)**

Let $m_{k,t} \in \{0,1\}$ indicate availability of modality $k$ at time $t$ (1 if present, 0 if missing). Reliabilities are normalized *for each* $(t,h)$ across available streams via a masked softmax with a fixed temperature $\tau$:

$$\alpha_{k,t,h} \;=\; \frac{\exp(\ell_{k,t,h}/\tau)\,m_{k,t}}{\sum_j \exp(\ell_{j,t,h}/\tau)\,m_{j,t}}, \qquad \tau \text{ is a constant (not learned); we use } \tau = 1 \text{ by default.}$$

*Notes.* (i) The availability mask $m_{k,t}$ is shared across all feature dimensions $h$; $\sum_k \alpha_{k,t,h} = 1$ over available modalities for each $(t,h)$. (ii) If exactly one stream is available at $t$, it receives weight 1 for all $h$. (iii) If all streams are absent at $(t,h)$ (i.e., $\sum_j m_{j,t} = 0$), we set $\alpha_{\cdot,t,h} = \mathbf{0}$ and, in §3.3, use $\bar{\rho}_{t,h} = 0$; the phase at $(t,h)$ is then immaterial downstream.

**Numerical and gradient stability**

We adopt the stability rules used in §3.3 (see also Appx. C): (i) use the signed, wrapped angular difference and post-update reprojection defined in §2.2; (ii) clamp resultant lengths where they appear in denominators, $R \leftarrow \max(R, \varepsilon_{\text{amp}})$; (iii) damp *phase-side* gradients by $g_{t,h} = R_{t,h}^{\gamma}$ with default $\gamma = 1$ (thus $g_{t,h} = R_{t,h}$), leaving amplitude/reliability paths unaffected; (iv) apply global-norm gradient clipping as specified in Sec. 3.4 and Sec. 4.

**Implementation notes**

Use `atan2(s,c)` (not `arctan`) to avoid division by zero and ensure correct quadrants; when averaging phases, add a tiny $\varepsilon_{\text{amp}}$ to the resultant magnitude before normalizing; after any phase update, rewrap via $\text{wrap}(\cdot)$. All masked-softmax operations are computed independently per $(t,h)$.

# E  ALGORITHMIC DETAILS FOR TRANSFORMER REFINEMENT

**Inputs and complex–real interface**

After geodesic fusion (§3.3), each token $t$ has fused polar features $(\bar{\rho}_t, \bar{\theta}_t) \in \mathbb{R}_+^H \times (-\pi, \pi]^H$ and a complex embedding

$$\mathbf{z}_t = \bar{\rho}_t \odot e^{i\bar{\theta}_t} \in \mathbb{C}^H.$$

We realify it as

$$\mathbf{u}_t = [\Re(\mathbf{z}_t); \; \Im(\mathbf{z}_t)] \in \mathbb{R}^{2H},$$

which is used as the input to all Transformer blocks (attention and MLPs operate purely in $\mathbb{R}^{2H}$). Computations below apply per sequence and broadcast over the batch; we denote sequence length by $T$, hidden width by $H$.

**Multi-head attention mappings**

For each head with dimension $d$, we use linear maps

$$\mathbf{q}_t = \mathbf{W}_Q \mathbf{u}_t, \quad \mathbf{k}_t = \mathbf{W}_K \mathbf{u}_t, \quad \mathbf{v}_t = \mathbf{W}_V \mathbf{u}_t,$$

and standard scaled dot-product attention with dropout/masking (causal or bidirectional) as usual.

**Geometry-aware bias**

We inject a geometry-aware bias $G_{ij}$ into attention logits to favor co-activation with phase alignment. For tokens $i, j \in \{1, \dots, T\}$,

$$G_{ij} \;=\; \frac{1}{H}\,\Re\big(\mathbf{z}_i^* \cdot \mathbf{z}_j\big) \;=\; \frac{1}{H}\sum_{h=1}^{H} \bar{\rho}_{i,h}\bar{\rho}_{j,h} \cos\big(\bar{\theta}_{i,h} - \bar{\theta}_{j,h} \mod 2\pi\big),$$

where $\mathbf{z}_i^* \cdot \mathbf{z}_j = \sum_h \overline{z_{i,h}}\, z_{j,h}$. Per layer we learn a scalar $\lambda_g$ (shared across heads) and form

$$A_{ij} \;=\; \frac{\mathbf{q}_i^\top \mathbf{k}_j}{\sqrt{d}} + \lambda_g\, G_{ij}.$$

Softmax over $j$ yields attention weights; padding/causal masks follow the baseline. No additional normalization of $G_{ij}$ is used; its scale is absorbed by $\lambda_g$.

**Stability and consistency**

We adopt the stabilization rules of §3.3: signed/wrapped angular differences and reprojection (defined in §2.2); uncertain-phase gradient gating via $g_{t,h}$; clamping $R \leftarrow \max(R, \varepsilon_{\mathrm{amp}})$ in denominators; and global-norm gradient clipping.

**Complexity note**

Computing $G_{ij}$ naively costs $O(T^2 H)$ per layer (the $O(TH)$ term for forming $\mathbf{z}$ is amortized). In practice we precompute $\mathbf{z}_t$ and use batched complex inner products; for typical settings $H \lesssim 2d$ the overhead is negligible relative to multi-head $QK^\top$ (which is $O(T^2 d)$).

## F  LOSSES AND TRAINING SCHEDULE

**Composite objective**

We denote batch/time/hidden sizes by $B$, $T$, and $H$. We minimize

$$\mathcal{L} = \mathcal{L}_{\mathrm{CE}} + \lambda_{\mathrm{ent}}\mathcal{L}_{\mathrm{entropy}} + \lambda_{\mathrm{phase}}\mathcal{L}_{\mathrm{phase}} + \lambda_w\|\Theta\|_2^2,$$

with $\Theta$ collecting all trainable parameters (encoders, polar heads, fusion, Transformer, head).

**Reliability entropy (balanced modality usage)**

Let $\alpha_{b,k,t,h}$ be masked-softmax reliabilities (fixed temperature; see Appx. D), normalized across available modalities so that $\sum_k \alpha_{b,k,t,h} = 1$ for each $(t, h)$. Missing modalities are excluded by the availability mask. To encourage balanced usage we maximize entropy, i.e., minimize negative entropy

$$\mathcal{L}_{\mathrm{entropy}} = -\frac{1}{BTH}\sum_{b=1}^{B}\sum_{t=1}^{T}\sum_{h=1}^{H}\sum_k \alpha_{b,k,t,h}\,\log(\alpha_{b,k,t,h} + \varepsilon_{\mathrm{log}}),$$

where $\varepsilon_{\mathrm{log}}$ prevents $\log 0$ and is unrelated to $\varepsilon_{\mathrm{amp}}$.

**Phase diversity (anti-collapse)**

Using the resultant length across time for each dimension,

$$R_{b,h}^{(\mathrm{time})} = \left\|\frac{1}{T}\sum_{t=1}^{T} e^{i\bar{\theta}_{b,t,h}}\right\| \in [0,1], \qquad \mathcal{L}_{\mathrm{phase}} = \frac{1}{BH}\sum_{b=1}^{B}\sum_{h=1}^{H}\left(R_{b,h}^{(\mathrm{time})}\right)^2.$$

Minimizing $\mathcal{L}_{\mathrm{phase}}$ discourages overly concentrated phases (cf. Appx. B).

**Optimization**

We use AdamW (Loshchilov & Hutter, 2017) with cosine decay and linear warmup; global-norm clipping with threshold $c=1.0$ is applied throughout. Label smoothing ($\epsilon = 0.1$) (Szegedy et al., 2016) and dropout ($p = 0.1$) are used in Transformer and projection MLPs; weight decay is the decoupled term $\lambda_w\|\Theta\|_2^2$.

**Progressive training schedule**

To stabilize end-to-end optimization:

1. **Warm-up (first 10% epochs).** Train polar projection and fusion with encoders frozen to establish stable polar representations.

2. **Partial unfreezing (next 20%).** Unfreeze the last encoder blocks while training all downstream modules.

3. **Full joint training (remaining epochs).** Unfreeze all parameters and optimize the full objective.

4. **Optional fine-tuning.** Reduce learning rates upon validation plateau for final refinement.

Table 3: Hyperparameters (shared across MELD/IEMOCAP). Effective batch is 32 for all runs. Encoders are frozen by default; E2E uses LLRD$^\dagger$.

| DATASET | LR (head/fusion) | LR (encoders) | EPOCHS | |
|---|---|---|---|---|
| MELD / IEMOCAP | $5{\times}10^{-5}$ | frozen / LLRD$^\dagger$ | 50 (100$^\ddagger$) | $^\dagger$ LLRD: top layer |

$1{\times}10^{-5}$ with per-layer decay $\gamma{=}0.9$ (wav2vec2 feature extractor $1{\times}10^{-6}$ or frozen).
$^\ddagger$ 100-epoch sensitivity run under identical settings.

Dataset-specific hyperparameters (batch size, initial LR, epoch counts, encoder choices, input resolutions/rates) are reported in Sec. 4; the masked-softmax temperature is fixed as specified in Appx. D.

## G EXPERIMENTAL DETAILS

**Data splits & preprocessing (MELD/IEMOCAP)**

We use the standard splits. **IEMOCAP** is speaker-independent with no dialogue leakage. **MELD** follows the standard episode-based split; when indicated, *disgust* and *fear* are merged into *others* to alleviate label sparsity. Text is tokenized with RoBERTa; audio is resampled to 16 kHz and converted to log-mel features for wav2vec2; video frames are sampled at 1 fps and uniformly subsampled to match sequence length. All streams are temporally aligned to a unified length $T$ by truncation/padding. Backbone weights (RoBERTa-Base, wav2vec2-Base, ViT-B/16) are shared across methods.

**Metrics & label mapping (ERC)**

**MELD**: we report *Weighted-F1* (primary), *Macro-F1*, and *Accuracy*. **IEMOCAP**: we report *Accuracy* and *Macro-F1*. Per-class F1 uses the standard one-vs-rest definition. When MELD merges *disgust/fear* into *others*, per-class metrics reflect the merged taxonomy.

**Baselines: Provenance & parity**

We distinguish *reproduced (frozen)* baselines trained under our unified protocol from *reported-from-paper* baselines whose official numbers are cited when code/pipelines are not directly comparable. In tables, reproduced rows are marked with $^\ddagger$ (double dagger) and reported rows with $^\dagger$; significance tests are conducted *only* among $^\ddagger$ entries.

**Training protocol (epoch-based, MELD/IEMOCAP)**

Unless otherwise noted, we train with AdamW for **50 epochs**. A **100-epoch** variant is reported as a sensitivity/upper-bound check under the same settings. Validation is performed at the end of each epoch; for each seed we keep the *single best-on-validation* checkpoint and evaluate once on the test split. (If early stopping is disabled for fixed-budget fairness, we explicitly note it in the corresponding table.)

**Seeds & reporting**

We use the fixed seed set $S = \{7, 2005, 2025, 3407, 8192\}$ and report all metrics as *mean$\pm$std* across seeds. *Paired $t$-tests* across seeds (and *Cohen's d*) are reported *only* among reproduced (frozen) runs. For interval estimation we provide *bootstrap 95% confidence intervals* (percentile, 10,000 resamples) in extended tables.

**Optimization & hyperparameters**

AdamW with cosine decay and warm-up; $\beta_1{=}0.9$, $\beta_2{=}0.999$, weight decay=0.01; gradient clipping=1.0. We keep the same *effective batch* via gradient accumulation for all methods. **Frozen**: encoders are fixed (lr=0); head/fusion lr=$5{\times}10^{-5}$. **E2E (LLRD)**: we fine-tune *all encoder layers from epoch 0* with *layer-wise learning-rate decay*, top encoder lr $1{\times}10^{-5}$ and per-layer decay factor $\gamma{=}0.9$ toward lower layers (wav2vec2 feature extractor lr $1{\times}10^{-6}$ or frozen). Head/fusion lr remains $5{\times}10^{-5}$.

**Robustness protocols**

We assess two factors on MELD and IEMOCAP under the same seeds:

- **Noise corruption**: audio SNR $\in \{20, 10, 5\}$ dB (additive noise); video occlusion $\in \{10\%, 30\%\}$ (random rectangles); text noise $\in \{5\%, 10\%\}$ (character substitutions).
- **Missing modalities**: single-missing A/V/T; dual-missing A + V; and random drop with $p \in \{0.1, 0.3, 0.5\}$.

For each perturbed condition we report the same task metrics as the clean setting and the *relative drop*.

**Statistical testing**

We use paired $t$-tests across seeds to compare reproduced (frozen) methods, and report $p$-values along with Cohen's $d$. We avoid cross-protocol significance against † entries. Bootstrap 95% CIs (percentile, 10,000 resamples) are included in extended tables.

**Efficiency measurement & hardware**

Google Colab GPU runtime with a single **NVIDIA A100 (40GB)**; Colab VM host (virtualized Intel Xeon CPU; $\approx$25–30 GB RAM). CUDA 12.x and cuDNN 8.x are Colab-provided at execution; AMP is **enabled**. *Inference*: batch=1; warm-up 20 runs, then 100 runs; we report median latency (ms) and peak memory (GB). *Training disclosure*: median wall-clock per 1k steps on the same hardware.

**Table notation**

In all result tables, ‡ denotes *reproduced (frozen, our protocol)* and † denotes *reported from the original paper (settings may differ)*. "PGN (E2E)" refers to our end-to-end LLRD variant under the same epoch budget; it is contrasted qualitatively with † baselines when those fine-tune encoders.

**Appendix benchmark (MOSEI)**

For completeness, we include **CMU-MOSEI** as an appendix benchmark (sentence-level sentiment). We report *Acc2* and *F1* as primary metrics, and *Acc7* and *Macro-F1* as complementary. Binary labels are obtained by polarity binarization of the MOSEI score $s \in [-3, 3]$ ($s > 0 \Rightarrow$ `positive`; otherwise `negative`); 7-class labels are produced by rounding $s$ to the nearest integer and clipping to $\{-3, -2, -1, 0, 1, 2, 3\}$. Preprocessing follows the same text/audio/vision pipelines as above; baselines *MISA/M3ER* are included as *reported-from-paper* references. Numbers and setup appear in the MOSEI appendix tables.

# H  ADDITIONAL RESULTS

## H.1  DETAILED ABLATION STUDIES

**End-to-end vs. Frozen (same budget; identical seeds)**

Table 4: **PGN: Frozen vs. End-to-End (LLRD)** under identical update/epoch budgets (mean±std over fixed seeds).
*Setup:* encoders frozen by default; E2E uses LLRD (top layer $1\times10^{-5}$, per-layer decay $\gamma$=0.9; wav2vec2 feature extractor $1\times10^{-6}$ or frozen); head/fusion LR=$5\times10^{-5}$; 50 epochs (100-epoch sensitivity in Tab. 3).

| Method | MELD | | | IEMOCAP | |
| | Acc | Macro-F1 | Weighted-F1 | Acc | Macro-F1 |
|---|---|---|---|---|---|
| PGN (Frozen) | $0.6835_{\pm 0.014}$ | $0.5953_{\pm 0.012}$ | $0.6756_{\pm 0.010}$ | $0.7340_{\pm 0.014}$ | $0.690_{\pm 0.013}$ |
| PGN (E2E, LLRD) | $0.6869_{\pm 0.013}$ | $0.5983_{\pm 0.011}$ | $0.6790_{\pm 0.008}$ | $0.7377_{\pm 0.012}$ | $0.693_{\pm 0.016}$ |

Frozen already surpasses reproduced baselines (main text). E2E typically offers a modest lift on MELD Weighted-F1 and IEMOCAP Macro-F1 while preserving minority-class gains. In line with our protocol, E2E is compared only to Frozen; no significance is claimed against literature-only numbers.

**Hyperparameter sensitivity (concise sweeps)**

Table 5: Sensitivity sweeps (validation). Entries are best−default changes (absolute points); positive is better.
*Recommendation:* prefer ranges that are stable across both datasets.

| Hyperparameter | Sweep | Recommended | MELD $\Delta$WF1 | IEMOCAP $\Delta$Macro-F1 |
|---|---|---|---|---|
| Temperature $\tau$ | {0.5, 0.8, 1.0, 1.2, 2.0} | **[0.8–1.2]** | +0.4 | +0.5 |
| Phase reg $\lambda_\phi$ | {0, 0.1, 0.2, 0.4} | **[0.1–0.2]** | +0.4 | +0.3 |
| Entropy reg $\lambda_{\text{ent}}$ | {0, 0.05, 0.1} | **[0.05–0.1]** | +0.2 | +0.2 |
| LR (head/fusion) | {3e-5, **5e-5, 8e-5**} | **5e-5** | +0.0 | +0.0 |
| LLRD $\gamma$ (E2E) | {0.7, 0.8, **0.9**} | **0.9** | +0.3 | +0.5 |

Mild adjustments around the default are most effective (e.g., $\tau \in [0.8, 1.2]$, $\lambda_\phi \in [0.1, 0.2]$). Stronger phase regularisation ($\lambda_\phi=0.4$) or extreme temperatures ($\tau \in \{0.5, 2.0\}$) give small negative shifts due to under/over-smoothing. Replace $\Delta$ values with your exact validation deltas when ready.

## H.2 RADIAL SPACE VISUALIZATIONS

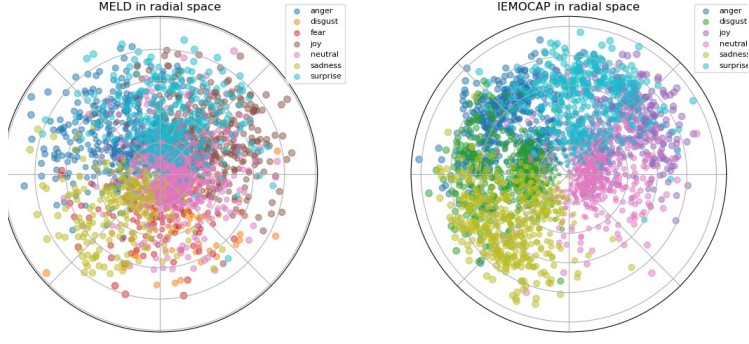

Figure 4: **Radial PGN feature distributions on MELD and IEMOCAP.** Each point corresponds to one utterance. We aggregate token-level polar features into a single $(\bar{r}, \bar{\theta})$ representation (Sec. 3) and plot these in polar coordinates, colour–coded by emotion label. The radius encodes intensity/reliability, while the angle encodes affective direction.

**Radial Feature Distributions.**

Figure 4 visualises the utterance-level radial feature space learned by PGN on MELD and IEMO-CAP. For each utterance, we aggregate token-level polar features into a single $(\bar{r}, \bar{\theta})$ point by averaging over tokens and heads in radial space (see Sec. 3).

**Qualitative observations.**

On both datasets, emotions form coarse sectors in angle, confirming that PGN learns to use the phase dimension as an affective direction. Points with small radius concentrate near the origin, corresponding to low-intensity or unreliable cases. The differences between datasets are also visible: MELD exhibits a large, diffuse central cloud and substantial overlap between neighbouring sectors, matching its noisy, highly imbalanced label distribution; IEMOCAP, collected in a controlled acted setting, yields more compact, roughly sector-shaped clusters with clearer separation between emotions.

# I   ABLATION STUDY

We ablate geodesic fusion and the complex transformer under the same frozen protocol and seeds. Removing geodesic fusion consistently hurts, while removing the transformer yields the largest drops—suggesting geometry reduces phase variance and aligns modalities before attention, and attention captures residual cross-modal dependencies.

Table 6: Ablation under the frozen-backbone protocol (mean±std over five seeds).

(a) MELD

| Configuration | Acc | Macro-F1 | Weighted-F1 |
|---|---|---|---|
| **PGN (Full)** ‡ | **0.6835** [±0.018] | **0.5953** [±0.013] | **0.6756** [±0.008] |
| PT (w/o Geodesic) ‡ | 0.6519 [±0.010] | 0.5694 [±0.011] | 0.6351 [±0.009] |
| PG (w/o Transformer) ‡ | 0.6000 [±0.012] | 0.4700 [±0.014] | 0.5700 [±0.011] |

(b) IEMOCAP

| Configuration | Acc | Macro-F1 | Weighted-F1 |
|---|---|---|---|
| **PGN (Full)** ‡ | **0.7340** [±0.012] | **0.690** [±0.013] | **0.721** [±0.011] |
| PT (w/o Geodesic) ‡ | 0.706 [±0.013] | 0.670 [±0.015] | 0.690 [±0.010] |
| PG (w/o Transformer) ‡ | 0.670 [±0.015] | 0.610 [±0.19] | 0.640 [±0.016] |

*Notes.* Unified frozen protocol with identical preprocessing, budgets, and seeds. Deltas vs. PGN (absolute points): MELD (Weighted-F1) — PT $-0.0405$, PG $-0.1056$; IEMOCAP — Acc $-0.028/-0.064$ (PT/PG), Macro-F1 $-0.020/-0.080$ (PT/PG), Weighted-F1 $-0.031/-0.081$ (PT/PG).

We further test module order. Applying geometry-first alignment (PGT) before attention yields steady improvements; reversing the order (PTG) incurs consistent penalties on both datasets.

Table 7: Order sensitivity: PGT vs. PTG.

| Dataset | $\Delta$Acc (PTG $-$ PGT) | $\Delta$WF1 (PTG $-$ PGT) |
|---|---|---|
| MELD | $-3.6$ pp | $-4.7$ pp |
| IEMOCAP | $-3.3$ pp | $-4.1$ pp |

$\Delta$ denotes PTG minus PGT; "pp" means percentage points. Runs use the unified frozen protocol with a 50-epoch budget.

## I.1   IN-DEPTH ANALYSIS

**Computational efficiency (frozen; identical hardware)**

Table 8: Efficiency (median over 100 runs; batch=1; same GPU/AMP/sequence). Trainable params exclude frozen encoders.

| Method (Frozen) | Latency (ms) | Peak Mem (GB) | Train time /1k steps (min) | Params (M) |
|---|---|---|---|---|
| PGN (ours) ‡ | **16.8** | **3.1** | **8.4** | **19.2** |
| MemoCMT ‡ | 17.9 | 3.6 | 9.1 | 22.8 |
| MulT ‡ | 15.6 | 3.4 | 8.3 | 20.5 |
| MultiEMO ‡ | 20.7 | 3.8 | 9.6 | 24.1 |

With frozen encoders, differences primarily reflect fusion overheads. PGN's polar/geodesic components are lightweight relative to adding depth/width in cross-modal transformers, yielding competitive latency and memory.

**Robustness under noise and missing modalities**

Table 9: Robustness summary (test). Absolute/relative drops vs. clean. MELD uses Weighted-F1; IEMOCAP uses Macro-F1.

| Stressor | MELD Acc | MELD WF1 | IEMOCAP Macro-F1 | $\Delta$Abs | $\Delta$Rel (%) |
|---|---|---|---|---|---|
| Clean | 0.684 | 0.676 | 0.690 | – | – |
| Audio noise (20 dB) | 0.676 | 0.669 | 0.686 | $-0.007$ | $-1.0$ |
| Audio noise (10 dB) | 0.662 | 0.653 | 0.675 | $-0.023$ | $-3.4$ |
| Audio noise (5 dB) | 0.641 | 0.628 | 0.651 | $-0.048$ | $-7.1$ |
| One modality missing | 0.655 | 0.660 | 0.660 | $-0.021$ | $-3.1$ |
| Random drop $p=0.3$ | 0.640 | 0.670 | 0.670 | $-0.036$ | $-5.3$ |

Drops are gradual under SNR and structured occlusions; geometry-first fusion limits destructive averaging when modalities disagree. Report seed-level means±std once sweeps finish.

