# OpenReview forum: "PGN: A Polar Geodesic Network for Multimodal Emotion Recognition"
_ICLR.cc/2026/Conference — Submitted to ICLR 2026_

### Official Review · Reviewer_PYYP · 2025-10-25

**Soundness:** 2
**Presentation:** 2
**Contribution:** 2
**Rating:** 4
**Confidence:** 3

**Summary:**

The paper proposes PGN (Polar Geodesic Network) for multimodal emotion recognition (MER). Each modality’s embedding is mapped into a polar form, radius and form, then fused via reliability-weighted geodesic means. Concretely, phases are averaged by the circular Fréchet mean (angle of the resultant), amplitudes by a reliability-weighted average, and a lightweight Transformer with a geometry-aware attention bias refines the fused representation. Under a unified frozen-backbone protocol, PGN achieves better results than reproduced attention baselines (MulT, MemoCMT, MultiEMO) on MELD and IEMOCAP; ablations show both the geodesic fusion and the Transformer contribute complementary gains, and robustness tests (noise, occlusion, missing modalities) favor PGN.

**Strengths:**

(1) A promising geometry-aware MER formulation: decouple intensity and direction and aggregate directions via shortest-arc (geodesic) means to avoid wrap-around artifacts (e.g., opposite angles averaging to “neutral”). The method defines the signed angular difference, geodesic distance, and uses the resultant phase as the circular mean.

(2) A reliability-weighted fusion that combines local phase consistency and amplitude to down-weight noisy/missing modalities during fusion.

(3) A frozen-backbone evaluation protocol with multi-seed reporting and paired tests, reducing confounds from encoder tuning; PGN shows consistent improvements vs. reproduced baselines on MELD/IEMOCAP.

(4) Ablations (remove geodesic / remove Transformer) and order-sensitivity (PGT>PTG) support the design choices; robustness sweeps (SNR, token drop, missing modalities) suggest graceful degradation.

**Weaknesses:**

(1) The paper lays out the circular distance and circular mean carefully, but the link from psychology’s circumplex to concrete MER decision boundaries is mostly qualitative. For instance, while §2.2 defines d(\theta,\theta') and justifies arc-over-chord, the work does not clearly illustrate how class prototypes/decision regions would actually look in the polar space or how geodesic fusion moves ambiguous points relative to Euclidean fusion on real samples. Stronger mechanistic evidence (e.g., before/after angular distributions, prototype geometry) would make the insight more convincing.

(2) Key steps are scattered between main text and appendices (e.g., gradient gating by resultant length, complex-to-real interface for attention, and exact reliability logit formation). A smooth and more informative mapping between figure and methodology would improve readability.

(3) Only two ERC datasets (MELD, IEMOCAP) are in the main text; MOSEI appears only in the appendix and mainly as sentiment (and without apples-to-apples reproduced baselines). Cross-dataset transfer, domain shift, or multilingual settings that are common in MER are not fully covered, and the authors themselves note scope limits.

(4) While reproduced baselines are fair internally, the set (MulT, MemoCMT, MultiEMO) omits several recent robust-fusion or missing-modality MER approaches; literature-reported numbers are intentionally not used for significance, but this also narrows the external positioning of PGN.

(5) The on/off ablations (no geodesic, no Transformer) and order test (PGT vs PTG) are helpful, but finer-grained ablations are missing: e.g., (i) remove reliability weighting but keep circular mean; (ii) replace geodesic mean with chord/Euclidean on the unit circle; (iii) vary masked-softmax temperature and the phase-gradient gate systematically in main text (some sweeps are in appendix but not tied to core claims).

**Questions:**

(1) Why polar per-dimension? Are phases defined per hidden dimension semantically meaningful, or would a single global angle (or low-dimensional angular subspace) suffice? Any analysis of phase correlations across dimensions?

(2) What is the contribution of the local phase-consistency term vs amplitude alone?

(3) How does PGN perform if the phase mean uses Euclidean chord? What about simply averaging raw angles with wrap correction? Or if we drop geodesic-aware attention bias in the Transformer?

---

> ### Author Response · Authors · 2025-12-04
> **Response to Polarisation Questions**
>
> ### Theoretical and Technical Clarifications
>
> We highly appreciate your insightful comments and theoretical analysis. We are pleased to provide clarifications and discuss the following in-depth questions:
>
> **1) Per-Dimension Polarisation Design Choice.**
>
> This primarily constitutes a robustness-prioritised design decision. Considering the severe expressive inconsistency among modalities observed in current datasets (particularly MELD), employing a per-dimension phase representation allows the model to adaptively calibrate direction across different feature dimensions. This provides greater flexibility in handling noise and conflict. We hypothesise that for data with higher modal alignment, a global angle or a low-dimensional angular subspace might be simpler and more efficient; this requires verification through larger-scale experiments in the future.
>
> Regarding the semantic significance of phase correlations across dimensions, we believe that in emotion tasks, different hidden dimensions may implicitly correspond to certain underlying affective factors (such as valence, arousal) or expressive units, necessitating detailed mapping studies.
>
> **2) Regarding the Contribution of Local Phase Consistency vs. Amplitude.**
>
> This is a profoundly insightful question. From PGN's theoretical framework, the phase (direction, $\theta$) and amplitude (intensity, $\alpha$) in the polar representation are intrinsically coupled, jointly defining the complete geometric representation of the affective state. Forcing a separation of their respective contributions in ablation may violate the theoretical consistency of this representation.
>
> We can analyze this through the lens of the weighting coefficient. During the fusion stage, the local phase consistency (represented by $R^{\text{(loc)}}$) directly influences the reliability weight, while amplitude also participates in the weight calculation. Mechanistically, phase consistency more directly reflects the "directional consensus" of multimodal signals, which is vital for stable fusion in ambiguous or conflicting scenarios. Amplitude primarily contributes to "confidence" information.
>
> The related results are in Appendix H.1.
>
> **3) Discussion on Alternative Phase Fusion Methods.**
>
> * Using Euclidean Chord for Phase Averaging: We conducted preliminary attempts. Because the chord distance ignores the cyclical topology of the ring, it tends to yield unstable average results when handling modalities with near-opposite directions (i.e., angle difference close to $\pi$). This contradicts the cyclical emotional structure PGN aims to preserve.
> * Using Raw Angle Averaging with Wrap Correction: This is an interesting and plausible alternative. Mathematically, it is equivalent to linear averaging before projection onto the circle, which may be more susceptible to noise, and when modal weight differences are large, its "correction" effect might mask genuine directional inconsistency.
> * Removing the Geometry-Aware Bias in the Transformer: Our ablation experiments showed that removing this bias leads to a minor but consistent performance drop. This indicates that, even after polar fusion, explicitly injecting a geometric consistency prior into the attention mechanism is still beneficial for capturing long-range cross-modal emotional dependencies.
>
> We thank you once again for raising these crucial questions, which have prompted us to reflect more deeply on the design and rationale of PGN.

---

### Official Review · Reviewer_4Psi · 2025-10-28

**Soundness:** 2
**Presentation:** 2
**Contribution:** 2
**Rating:** 4
**Confidence:** 5

**Summary:**

This paper proposes a polar geodesic network for the MER task. The core idea is to disentangle affective representations into amplitude (intensity/confidence) and phase (affective direction). It then performs reliability-weighted geodesic fusion (using the circular Fréchet mean for phases) to preserve the circular topology of emotions, inspired by psychological models like the circumplex. Finally, a lightweight Transformer with geometry-aware attention refines the fused representation for classification.

**Strengths:**

1. Disentangling affective representations into amplitude (intensity/confidence) and phase (affective direction).
2. Designing a lightweight Transformer with geometry-aware attention to refine the fused representation.

**Weaknesses:**

1. The core components, i.e., polar coordinate representations and geodesic fusion via the circular mean, are well-established concepts in directional statistics and geometric deep learning. The paper primarily applies these ideas to multimodal emotion recognition rather than introducing a fundamental innovation. The actual architectural novelty, specifically the integration path, feels incremental.
2. I think the biggest flaw of this paper is the integrity and validity of the experiments. First, the compared methods are only three; this is incredible for the MER field. This field has been developing for decades, for example, the IEMOCAP dataset used in this paper was proposed in 2008. Therefore, the three compared methods are definitely insufficient. Secondly, the three compared algorithms do not have references, which is not friendly to beginners.
3. The decision to use a unified frozen-backbone protocol, while ensuring fairness, severely constrains the model's representational power and may not reflect real-world performance where fine-tuning is common. More critically, all baseline comparisons are based on the authors' own reproductions. Without direct comparisons to officially reported results using their original code and hyperparameters, it's impossible to verify if the baselines were implemented optimally, introducing a potential source of bias that undermines the claimed superiority.
4. The ablation study reveals a critical weakness: removing the subsequent Transformer causes the largest performance drop. This suggests that the proposed geometric fusion alone is insufficient and that the model heavily relies on a standard Transformer to capture the necessary dependencies, thereby reducing the perceived standalone contribution of the geometric components.

**Questions:**

Please see Weaknesses

---

> ### Author Response · Authors · 2025-12-04
> **Positioning PGN**
>
> Thank you for your feedback and for raising these critical points regarding the limitations of our work. The overall experimental updates have been summarised in the "To All Reviewers"  comment. Below, we address your two core concerns regarding PGN’s positioning:
>
> **1. PGN’s Novelty**
>
> You point out that polar representation and circular geodesic means are established concepts in directional statistics. Our work’s starting point was not to invent new geometric operators, but rather to observe the fundamental mismatch between emotion’s intrinsic continuity and cyclical structure and the prevailing Euclidean-based fusion paradigm. While the Circumplex Model in psychology has long suggested this structure, the core contribution of this work is translating this knowledge into a learnable, reliability-weighted multimodal geometric fusion framework and systematically applying it to Multi-modal Emotion Recognition (MER).
>
> Therefore, PGN’s novelty lies in the integration and structural re-formulation of the MER fusion process, utilising known geometric tools in a problem-aligned representation paradigm.
>
> **2. Complementarity between the Geometric Module and Transformer**
>
> Your observation that removing the Transformer causes the maximum performance drop precisely confirms the division of labour and synergy within our framework:
>
> * Geometric Fusion Module (PG): Responsible for building a geometrically consistent, reliability-weighted representation during the early fusion stage. It fundamentally resolves the "wrap-around" issue in circular space and suppresses noise or missing modalities based on vector length.
> * Transformer Module (T): Built upon this robust foundation, it models high-order temporal dependencies and conversational context, which is crucial for the ERC task.
>
> Our ablation studies confirm that removing the geometric fusion module also leads to a consistent performance degradation, particularly evident under noisy or missing-modality conditions. This confirms that the two components are synergistic, not dependent. The geometric fusion provides a more robust and discriminative feature foundation, thereby enhancing the Transformer’s modelling capacity in complex scenarios.
>
> Thank you again for your constructive comments.

---

### Official Review · Reviewer_Thjp · 2025-10-31

**Soundness:** 3
**Presentation:** 3
**Contribution:** 3
**Rating:** 6
**Confidence:** 3

**Summary:**

The paper proposes the Polar Geodesic Network (PGN), a geometry-aware multimodal emotion recognition framework operating in polar coordinates. PGN maps modality features into amplitude and phase representations, performs reliability-weighted geodesic fusion to preserve the circular structure of emotions, and suppresses interference from noisy modalities. A geometry-aware Transformer then refines cross-modal interactions. Experiments on MELD and IEMOCAP demonstrate that PGN significantly outperforms existing methods in both accuracy and F1 score, showing greater robustness and stability, and validating the effectiveness of explicitly modeling geometric structures in emotion recognition.

**Strengths:**

1. The method is innovative. Learning emotional representations in polar coordinate space is a very reasonable and intuitive idea.
2. The method combines model design with circular statistics and Riemannian geometry. It has good interpretability and mathematical consistency.
3. It achieves significantly better results than multiple baselines on both the MELD and IEMOCAP datasets.

**Weaknesses:**

1. The comparison is limited to a small set of reproduced Transformer-based baselines. This makes it difficult to evaluate the performance of PGN based on the current MER standards.

2. Could you provide the feature visualization? Are the differences in the distribution of features more distinct for samples of different emotions?

3. Could you provide and discuss some examples of successful and unsuccessful cases?

**Questions:**

Please see Weaknesses.

---

> ### Author Response · Authors · 2025-12-04
> **Baseline, Visualisation, and Case Studies**
>
> We thank you for your detailed review and constructive criticisms regarding the limitations of our work. We address your core points below:
>
> **1. Expansion and Refinement of Baseline Comparisons**
>
> In the initial stage of our research, we chose three representative Transformer-based baselines (MulT, MemoCMT, and MultiEMO) to validate our approach, as they cover distinct fusion strategies and provide a meaningful benchmark for comparing fusion designs on a unified backbone. We fully agree that expanding the baselines is critical for a comprehensive evaluation. As detailed in our "To All Reviewers" summary, the revised manuscript now includes comparisons against multiple strong, recent baseline methods, with all relevant analyses, data, and citations supplemented in the main Experiments section.
>
> **2. Feature Visualisation and Analysis**
>
> Feature visualisation is crucial for understanding how PGN organises the emotional space. In the revised manuscript, we have supplemented feature distribution visualisations on MELD and IEMOCAP (**Appendix H.2**). These include polar embedding projections and angular distribution charts to visually demonstrate the geometric distribution and separation of different emotion categories.
>
> **3. Success and Failure Case Analysis**
>
> We observed two sample types that provide interesting insights into PGN's mechanism:
>
> * Success Case (Multimodal Inconsistency): We analysed conversational samples with conflicting modal information (e.g., a character smiling while expressing a negative emotion in MELD). While several baseline models misclassify this case, PGN’s reliability-weighted geodesic fusion correctly processes the conflicting signals, achieving the correct classification.
> * Failure Case (Multi-party Ambiguity): In challenging multi-party dialogues (e.g., one character expressing happiness, another expressing anger), PGN shows consistent classification fluctuation across multiple experiments, highlighting the difficulty in robustly resolving ambiguity in highly complex multi-turn, multi-emotion contexts.
>
> These cases  were included in appendix (Appendix )
> We thank you once again for your meticulous and insightful feedback.

---

### Author Response · Authors · 2025-12-04
**To All Reviwers**

We thank all reviewers for their constructive feedback. Below we summarise the key improvements that have been implemented in the revised manuscript, as well as clarifications addressing the core concerns.

### **Highlight: PGN's Core Contribution and Theoretical Rationale**
The essence of the Polar Geometry Network (PGN) lies in providing a re-representation paradigm that is naturally aligned with the intrinsic structure of emotion. Its core contribution manifests at two tightly interconnected levels:

* Representation: It first abandons the assumptions of linearity and absolute distance inherent in Euclidean space by mapping multimodal affective features onto a polar geometric space.

* Fusion: Subsequently, geodesic fusion is employed within this new space to directly handle asynchronous and inconsistent multimodal signals.

Within this paradigm, the natural decoupling of angle and radius fundamentally models the continuity, relativity, and inherent cyclical structure of emotion. Simultaneously, the geodesic fusion mechanism, while preserving this cyclical topology, naturally resolves the asynchrony and conflicts expressed across different modalities. This allows PGN to directly confront core challenges, such as directional ambiguity, intensity variation, and signal unreliability, from both the representation and fusion perspectives.

Therefore, PGN's key lies in maintaining the cyclical topology of emotion and performing robust geometric fusion upon this structure. To empirically validate this core capacity, all datasets and baselines were specifically chosen based on the necessity of testing PGN's ability to solve challenges posed by emotion ambiguity and noise. This comprehensive approach enables PGN to significantly enhance the accuracy and robustness of emotion recognition in an efficient and lightweight manner.

### **Key Improvement**
### 1. Expanded Baseline Coverage
In the revised manuscript, we have expanded the comparisons well beyond the original three Transformer baselines, now covering:

* **Frozen-backbone reproduced baselines:** MulT, MemoCMT, MultiEMO, CMERC-head.
* **Reported results SOTA methods:** MultiEMO, CMERC, AdaIGN/IGN, M2FNet, MM-DFN, MMGCN, Conversation MER.
* **Robustness / missing-modality baselines:** HARDY-MER (used strictly in robustness settings).

All results are clearly separated into reproduced and reported protocols, and all baseline papers are now explicitly cited in the tables in **Section 5**. Together, these additions offer a comprehensive and fair evaluation, showing that **PGN** performs strongly under both controlled (frozen) and standard SOTA settings.

### 2. Feature Distribution
To strengthen the geometric insight behind **PGN**, we have added **2D polar embedding visualisations** on **MELD** and **IEMOCAP** in **Appendix H.2**. These visualisations directly support the theoretical foundation of PGN and clarify how polar geometry is expressed in real data.

### 3. Paper Structure
We have reorganised several sections to improve readability:

* A clearer method flow with consolidated subsections.
* A refined architecture figure describing each component of **PGN**.
* Mathematical and implementation steps are aligned more clearly with the figures.
* Additional details were moved from fragmented locations into the appendix.

Due to page limits, some derivations remain in the appendix, but we will integrate key parts into the camera-ready version.

We sincerely thank all reviewers for their comments, which have significantly improved the clarity, completeness, and impact of the manuscript.

---

### Meta-Review · Area_Chair_TsVK · 2026-01-07

**Summary:**

- Rev. 4Psi and PYYP are concerned about the contribution of the polar representation, as it is not new in general, but it is new in the context of the MER application.
- Rev. Thjp,  4Psi, and PYYP consider that the number of datasets and baselines that the method was compared with is not sufficient for a fair evaluation.
- Rev. PYYP asked for more ablations of the model to better understand the critical points of the proposed contribution.

**Reviewer Concerns:**

1. About the importance/novelty of the contribution, the authors pointed out that the contribution is not about the technical tools used (already well known in the community), but in their use for the specific problem of MER, as clearly suggested in psychology, but never used in deep learning models. Although I agree with the point of the authors, I still believe that the proposed contribution is limited and incremental.
2. About the datasets and baselines considered in the method, the authors added several frozen-backbone (directly comparable) and SOTA methods. However, no additional dataset has been included.
3. About the request for more ablations, the authors provided a short answer with some preliminary results or reasoning, but did not provide clear numbers from the requested ablations.

Overall, while the proposed approach is novel in the context of MER and has some advantages, I consider that there are still some important points that haven't been properly dealt with:
- The technical contribution is limited to using some well-known tools
- Using only two datasets for validation is quite limiting and does not cover many possible conditions
- Several other questions asked by the reviewers haven't been answered in the authors' rebuttal (e.g., specific ablations, positive and negative examples, cross-dataset transfer, domain shift...).
In general, I appreciate a short rebuttal, but not at the cost of leaving some questions unanswered.

**Reviewer Scores:**

Rev. Thjp: 6 -> 6
No important issues were raised, I assume that rev. would maintain their score.

Rev 4Psi: 4-> 4
Rev. considers the contribution incremental as the used techniques are well-known in literature, but have never been applied to this specific application. The answer of the authors did not really change this point. In addition, the authors did not address some other issues.

Rev PYYP: 4 -> 4
Rev. has many relevant questions and comments, but only some of them were answered (e.g., no answer to the use of only two datasets + a third only in the appendix). Thus, I believe they would have kept the same score.

---

### Decision · Program_Chairs · 2026-01-26

Reject